# MULTI-VIEW MASKED AUTOENCODERS FOR VISUAL CONTROL

## ABSTRACT

This paper investigates how to leverage data from multiple cameras to learn representations beneficial for visual control. To this end, we present the Multi-View Masked Autoencoder (MV-MAE), a simple and scalable framework for multi-view representation learning. Our main idea is to mask multiple viewpoints from video frames at random and train a video autoencoder to reconstruct pixels of both masked and unmasked viewpoints. This allows the model to learn representations that capture useful information of the current viewpoint but also the cross-view information from different viewpoints. We evaluate MV-MAE on challenging RLBench visual manipulation tasks by training a reinforcement learning agent on top of frozen representations. Our experiments demonstrate that MV-MAE significantly outperforms other multi-view representation learning approaches. Moreover, we show that the number of cameras can differ between the representation learning phase and the behavior learning phase. By training a single-view control agent on top of multi-view representations from MV-MAE, we achieve 62.3% success rate while the single-view representation learning baseline achieves 42.3%.

## 1 INTRODUCTION

Recent self-supervised learning approaches have been successful at learning useful representations from multiple views of the data, including different channels (Zhang et al., 2017) or patches (Oord et al., 2018) of an image, vision-sound modalities (Owens et al., 2016), vision-language modalities (Radford et al., 2021; Alayrac et al., 2022), and frames of a video (Wang & Gupta, 2015). The main underlying idea of these approaches is to utilize information about the same data from different perspectives as supervision for representation learning. Notably, Zhang et al. (2017) trained an autoencoder that predicts a subset of the image channels from another subset, and Radford et al. (2021) trained a vision-language model that matches image-text pairs with contrastive learning. Promising results from these approaches suggest that data diversity can play a key role in representation learning.

In the context of visual control, the camera is an easily accessible instrument that can increase data diversity by providing information about the same scene from different viewpoints. For instance, it has been a widely-used technique for roboticists to utilize multiple cameras for solving complex manipulation tasks (Akkaya et al., 2019; Akinola et al., 2020; Hsu et al., 2022; James et al., 2022; Jangir et al., 2022). Yet these works mostly focus on the improved performance from utilizing multi-view observations as inputs, not investigating the effectiveness of representation learning with diverse data from multiple cameras. A notable exception is the work from Sermanet et al. (2018), which learns view-invariant representations via contrastive learning. However, enforcing viewpoint invariance assumes that all viewpoints share similar information and thus requires careful curation of positive and negative pairs, similar to other contrastive approaches that often depend on complex design choices about sampling such pairs (Arora et al., 2019).

We present Multi-View Masked Autoencoders (MV-MAE), a simple and scalable framework for visual representation learning with diverse data from multiple cameras. Our main idea is to mask randomly selected viewpoints and train an autoencoder that reconstructs pixels of both masked and unmasked viewpoints. This allows the model to learn representations of each viewpoint that captures visual information of the current viewpoint but also cross-view information of other viewpoints. To further encourage cross-view representation learning, we propose to train a video autoencoder by masking multiple viewpoints from video frames at random. Because the model can utilize information

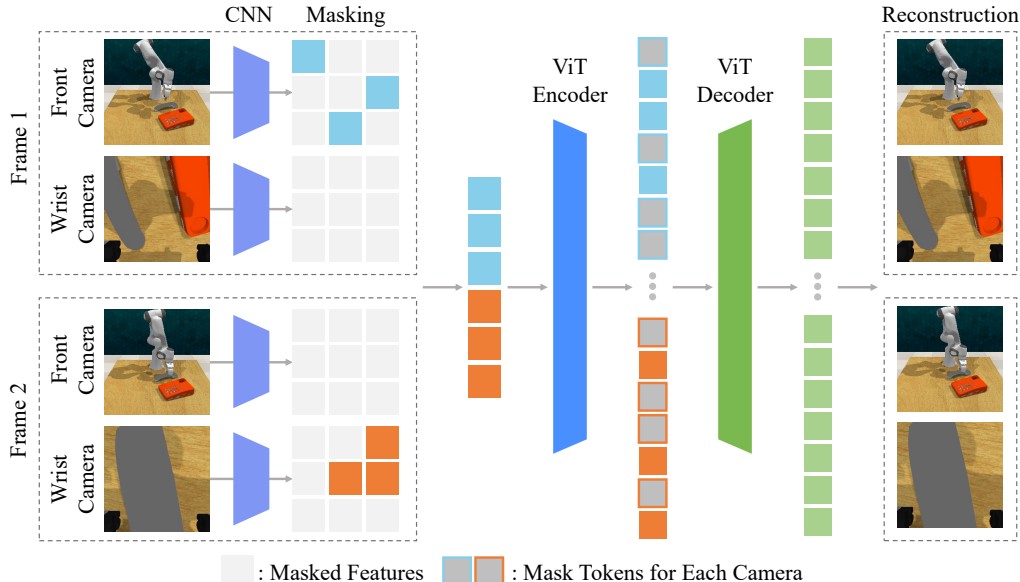

Figure 1: Illustration of our representation learning scheme. We extract features from each viewpoint with convolutional networks (CNN) and mask all features from randomly selected viewpoints of video frames. We also mask randomly selected features from remaining viewpoints to encourage the autoencoder to learn information of unmasked frames. A vision transformer (ViT; Dosovitskiy et al. 2021) encoder processes visible features to fuse information from multiple views and frames. Then a ViT decoder concatenates mask tokens for each view and processes inputs to reconstruct frames. We note that the autoencoder reconstructs all frames at the same time.

from the current frame but also information from unmasked frames of the target view, we find our approach helps the model to focus on predicting important details, *e.g.,* gripper poses. Then we utilize learned representations for visual control by training a reinforcement learning agent that learns a world model on top of frozen representations and utilizes it for behavior learning (Seo et al., 2022a).

**Contributions.** We highlight the contributions of our paper below:

- We present MV-MAE, a simple and scalable framework that can leverage diverse data from multiple cameras for visual representation learning. The main idea of MV-MAE is training a *video masked autoencoder* (Tong et al., 2022; Feichtenhofer et al., 2022) with a *view-masking* strategy that encourages the model to learn spatial dependency between viewpoints.

- We provide empirical evaluation of MV-MAE on challenging visual manipulation tasks from RLBench (James et al., 2020). Unlike other multi-view representation learning baselines that enforce invariance between multiple viewpoints (Sermanet et al., 2018; Assran et al., 2022), MV-MAE consistently outperforms a single-view representation learning baseline under a challenging experimental setup with multiple cameras of diverse types.

- We demonstrate that data diversity from multiple viewpoints can play a crucial role in representation learning for visual control. In particular, by training a single-view control agent on top of multi-view representations from MV-MAE, we find that our approach significantly outperforms a single-view representation learning baseline, *e.g.,* we achieve 62.3% success rate on six visual manipulation tasks while the baseline achieves 42.3%.

## 2 RELATED WORK

**Unsupervised visual representation learning.** Self-supervised learning on large-scale unlabeled datasets has been actively studied in the domain of computer vision (Noroozi & Favaro, 2016; Chen et al., 2020; Grill et al., 2020; He et al., 2021). A line of research that has been successful is contrastive learning, which learns representations by maximizing the mutual information between

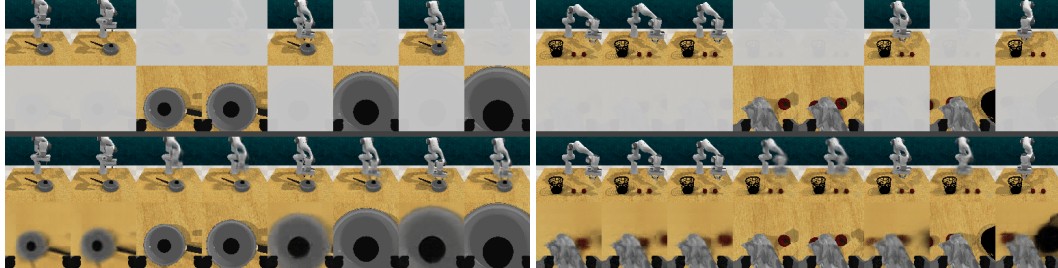

Figure 2: Masked view reconstruction on Take Lid Off Saucepan (left) and Put Rubbish in Bin (right) tasks from RLBench (James et al., 2020). We visualize ground-truth frames with masked viewpoints (upper two rows) and ground-truth frames with reconstructed frames (lower two rows). We find that the model can successfully reconstruct masked viewpoints.

different views of images (Chen et al., 2020; He et al., 2020; Tian et al., 2020a;b). The work closest to ours is Sermanet et al. (2018), which proposed to learn view-invariant representations with videos collected with multiple cameras. But contrastive approaches often require complex design choices with regard to sampling positive-negative pairs (Arora et al., 2019) and can fail when the assumption for induced invariance does not hold for downstream tasks (Xiao et al., 2021). We instead take a simple approach that learns representations via masked reconstruction.

Our work is closely related to denoising autoencoding approaches (Vincent et al., 2010; Pathak et al., 2016) that train an autoencoder to reconstruct masked inputs. Notably, He et al. (2021) proposed a scalable masked autoencoder architecture that masks a large portion of an image and only uses visible patches as inputs for solving reconstruction tasks. Our model is built on top of its spatiotemporal extension that introduces a video autoencoder (Tong et al., 2022; Feichtenhofer et al., 2022). This work follows this line of research by introducing the autoencoder that reconstructs masked viewpoints. Recently, Geng et al. (2022) trained a unified autoencoder for vision-language modalities with masked token prediction and found that data diversity allows for learning transferable representations. We support this finding with our investigation that shows data diversity from multiple cameras can be helpful for representation learning in the context of visual control.

**Unsupervised representation learning for visual control.** Representation learning from visual observations has also been actively studied for learning to solve control tasks from easily accessible cameras (Watter et al., 2015; Oord et al., 2018; Gelada et al., 2019; Hafner et al., 2019; Yarats et al., 2021b; Srinivas et al., 2020; Castro, 2020; Schwarzer et al., 2021a; Yarats et al., 2021a; Seo et al., 2022b). Recent works have demonstrated that self-supervised learning can enable agents to solve visual control tasks with frozen representations (Stooke et al., 2021; Schwarzer et al., 2021b; Nair et al., 2022; Parisi et al., 2022; Xiao et al., 2022; Seo et al., 2022a; Radosavovic et al., 2022). Our work further demonstrates that frozen representations learned with diverse data from multiple camera viewpoints can be effective for solving challenging visual manipulation tasks.

**Visual control with multiple cameras.** Leveraging multiple cameras has long been considered a practical and feasible technique in robotics, as the camera is usually an affordable and ubiquitous device (Sola et al., 2008; Carrera et al., 2011; Yang et al., 2021). Based on recent advances in computer vision and robot learning, there have been several approaches that utilize multi-view data from multiple cameras for visual control (Sermanet et al., 2018; Akinola et al., 2020; Zhan et al., 2020; Chen et al., 2021a; Hsu et al., 2022; Jangir et al., 2022; Shridhar et al., 2022; Guhur et al., 2022). While most approaches utilize multi-view data directly as inputs for robots, recent works have demonstrated that self-supervised learning that learns view-invariant representations (Sermanet et al., 2018) or 3D keypoints (Chen et al., 2021a) can be useful for downstream tasks. This work also demonstrates that multi-view representation learning can be beneficial for visual control.

## 3 Multi-View Masked Autoencoders for Visual Control

To fully exploit the multi-view data for representation learning, it is important to encourage the representations to learn cross-view information between viewpoints. In this section, we first introduce the

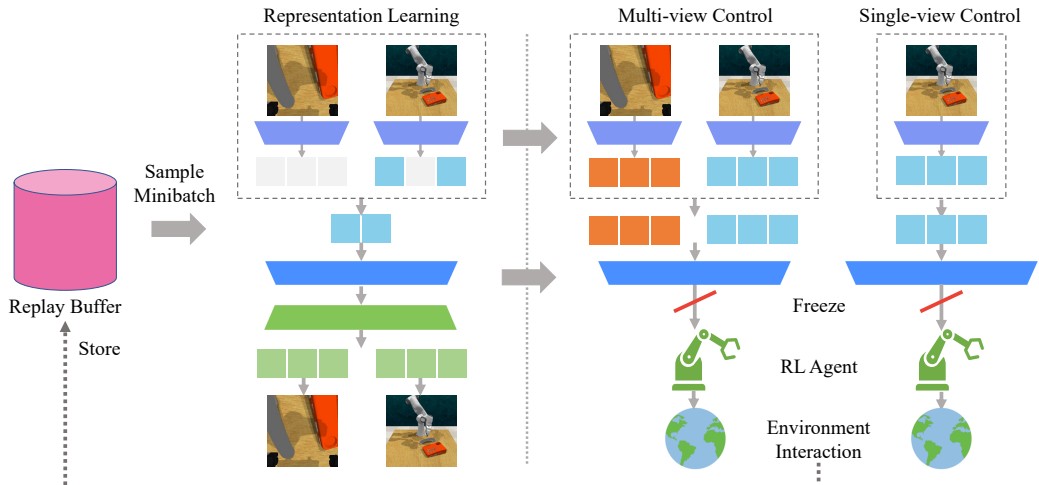

Figure 3: Overall pipeline of our framework. In the representation learning phase, we learn visual representation from multi-view data sampled from a replay buffer, as we have illustrated in Figure 1. In the control phase, we train an RL agent upon the frozen representations and store the samples collected from online interaction to the buffer. Thanks to our architecture design that can take inputs of varying lengths, MV-MAE can take inputs of either single-view or multi-view. This allows us to utilize a different number of cameras for the representation learning and behavior learning phases. We provide a pseudocode of our framework in Appendix B.

Multi-View Masked Autoencoder (MV-MAE) that reconstructs missing pixels of masked viewpoints for multi-view representation learning (see Section 3.1). We then describe how we utilize the visual representations for training a reinforcement learning (RL) agent (see Section 3.2).

## 3.1 MULTI-VIEW REPRESENTATION LEARNING

Our main idea for multi-view representation learning is to learn cross-view information from multiple viewpoints by reconstructing missing pixels of randomly masked viewpoints. However, such masked view reconstruction might be too challenging for the autoencoder without any access to information from missing viewpoints. For instance, reconstructing all details of a front camera observation, which contains a broader view of a robot workspace, with only having access to a wrist camera could be extremely challenging (see Figure 4 for examples of front and wrist cameras).

To address this issue, we propose to train a video masked autoencoder (Feichtenhofer et al., 2022; Tong et al., 2022) with *view-masking* that reconstructs missing pixels of randomly masked viewpoints from *video frames*. Because the autoencoder attends to unmasked neighbor frames from the same view, the model can focus on modeling important information such as target object positions and the movement of robot arms, while ignoring redundant information such as background for reconstructing masked viewpoints. We provide the overview of our representation learning scheme in Figure 1.

**Convolutional feature embedding.** Unlike prior work that masks random pixel patches (He et al., 2021; Feichtenhofer et al., 2022; Tong et al., 2022; Geng et al., 2022), we embed camera observations into convolutional feature maps following the design of Seo et al. (2022a).[1] Specifically, we downsample $96 \times 96 \times 3$ input images to convolutional feature maps with the spatial size of $6 \times 6$ by introducing 4 convolutional layers. Note that we separately process observations from each viewpoint with convolutional layers that share parameters. For each viewpoint, we add fixed 2D sin-cos position embeddings (Chen et al., 2021b) to the features. Then we add learnable 1D parameters representing

---

[1]Seo et al. (2022a) observed that masked image modeling with pixel patch masking (He et al., 2021) can make it difficult for the model to learn fine-grained details *within* patches, *e.g.,* object positions and boundaries, which could be crucial for tasks that require precise control such as visual manipulation tasks.

each camera and timestep to features of each video frame from different viewpoints. This is based on the idea of Geng et al. (2022) that introduces learnable parameters for vision and language inputs. Then we flatten the features and concatenate them into a single sequence.

**View masking.** We introduce a new view-masking scheme that masks all the features from a randomly selected camera viewpoint to encourage the model to learn cross-view information. Specifically, we mask randomly selected viewpoints from video frames by randomly sampling one viewpoint for each frame. We also mask randomly selected features from remaining viewpoints (see Figure 1) because we want the autoencoder to learn not only cross-view information but also the information within each viewpoint by reconstructing raw visual observations with masked features. We empirically find that the proposed view-masking can be more effective than uniform-masking scheme by explicitly encouraging multi-view representation learning (see Figure 7(a) for supporting experiments).

**Video autoencoding.** For autoencoder architecture, we largely follow the design of masked autoencoders (MAE; He et al. 2021; Feichtenhofer et al. 2022; Tong et al. 2022) that utilize the encoder and decoder consisting of vision transformers (ViT; Dosovitskiy et al. 2021). Specifically, the encoder processes a sequence of unmasked features from all viewpoints and video frames through a series of ViT layers. Then we concatenate a set of mask tokens with encoded features and add learnable parameters for each viewpoint and each time to corresponding features and mask tokens. The decoder processes them through ViT layers and linearly projects them into pixel patch predictions (see Figure 2 for examples of masked view reconstruction). The training loss for the autoencoder is the mean squared error between patch predictions and ground-truth pixel patches with a spatial size of $16 \times 16$.

## 3.2 VISUAL CONTROL

Once we learn multi-view representations, we train an RL agent on top of the representations for visual control. Because MV-MAE consists of ViTs based on a transformer architecture that can take inputs of variable lengths (Vaswani et al., 2017), the encoder can extract both single-view and multi-view representations though we utilize multi-view data for representation learning. This can be beneficial in practical scenarios where we can utilize additional third-person cameras during training, but the robot should operate on a fewer number of cameras at deployment time for faster inference.

**Reinforcement learning.** We consider masked world models (MWM; Seo et al. 2022a) as our base RL algorithm, which learns a world model on top of frozen MAE representations and utilizes it for behavior learning (Hafner et al., 2021). Because MWM decouples visual representation and dynamics learning, it naturally aligns with our approach of separately training the autoencoder to learn visual information from multiple cameras. We provide more details in Appendix C.

**Training.** Throughout the training, we iterate the processes of (i) updating MV-MAE with multi-view data, (ii) training the MWM agent with single-view or multi-view data, and (iii) collecting samples with environment interaction. As inputs to MWM agent, we concatenate the convolutional features with proprioceptive states and process them through the MV-MAE encoder. Because MWM is a model-based approach with a recurrent architecture, we extract representations from a single video frame for behavior learning. This is based on the observation of Feichtenhofer et al. (2022), where image representations from the video autoencoder are found to be useful for image recognition.

## 4 EXPERIMENTS

We design our experiments to investigate the following questions:

- Can MV-MAE effectively leverage multi-view data for representation learning?
- Can multi-view representation learning be useful for single-view visual control?
- Can MV-MAE work with more than two cameras?
- What is the effect of each component in MV-MAE?

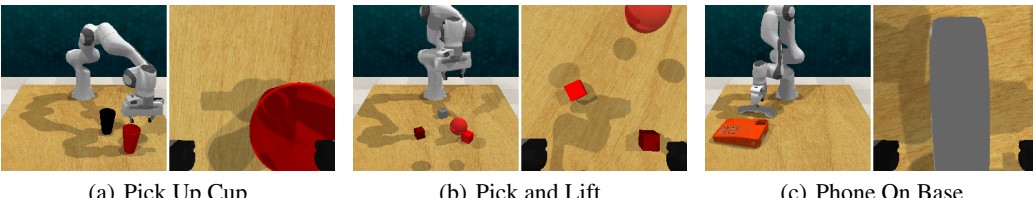

(a) Pick Up Cup        (b) Pick and Lift        (c) Phone On Base

Figure 4: Examples of multi-view data consisting front and wrist camera observations used in our RLBench (James et al., 2020) experiments. Front camera observations provide a broad look at a robot workspace, and wrist camera observations provide a closer look at target objects. We provide the examples of multi-view data from all six tasks in Appendix D.

## 4.1 SETUP

**Environments.** We investigate the effectiveness of representation learning with data from multiple viewpoints on challenging visual manipulation tasks from RLBench (James et al., 2020) — a standard benchmark for vision-based robotics which has been shown to serve as a proxy for real-robot experiments (James & Davison, 2022). While RLBench is originally designed to evaluate the performance in a sparse reward setup, we design dense rewards for six manipulation tasks. Moreover, to ease the difficulty of exploration in large action space, we enforce a robot gripper to be in an upright position without rotation. Following James & Davison (2022), we also fill a replay buffer with 100 expert demonstrations. We find these schemes allow us to focus on investigating the effect of visual representation learning while avoiding the challenge from sparse rewards and hard exploration. Unlike prior approaches that utilize path planner with the policy to output next best gripper pose (James & Davison, 2022; James et al., 2022), our RL agent outputs relative change in gripper position. We provide further details and source code for the environments in Appendix C.

**Implementation.** For all experiments, we use only 96×96 RGB observations from each camera. We downsample images by 16× to obtain convolutional feature maps with a spatial size of 6×6. Unlike prior approaches that train video masked autoencoders (Feichtenhofer et al., 2022; Tong et al., 2022), we do not utilize temporal downsampling because the agent operates on each image. For view-masking, we apply random view-masking independently to each frame in a video consisting of 4 frames. We compute the masking ratio in our experiments by including the number of features from masked viewpoints; hence 50% masking ratio means no features are masked from remaining viewpoints in a two-camera setup. Our autoencoder consists of the 8-layer ViT encoder and the 6-layer ViT decoder, where the embedding dimension is set to 256. We do not utilize a learning rate schedule as a replay buffer keeps receiving new samples. Following the design of Seo et al. (2022a), we also introduce a reward prediction objective for the autoencoder to encode task-relevant information into visual representations. We provide more details and source codes for reproducing our results in Appendix C.

**Baselines and our framework.** We consider following baselines for our experiments. For a fair comparison, we ensure that all methods use the same amount of data for representation learning. Specifically, for a single-view representation learning baseline, we train a single-view encoder on camera observations from both viewpoints. We also use the same architecture for all methods. More details on baselines are available in Appendix A.

- Masked world models (MWM; Seo et al. 2022a): MWM learns visual representations of a single viewpoint by training an image autoencoder and trains a world model on top of frozen representations. Because MWM does not encode cross-view information into representations, comparison with MWM evaluates the benefit of multi-view representation learning.

- MWM + Masked Siamese Network (MSN; Assran et al. 2022): MSN learns visual representations by matching the representations of masked images and the augmented images. We modify MSN to learn cross-view information by making it match the representation of the masked viewpoint with the unmasked viewpoint. We apply MSN to MWM and use it as a baseline. Comparison with MWM + MSN evaluates the benefit of our approach against representation learning that enforces view-invariance via positive pair matching (Grill et al., 2020; Assran et al., 2022).

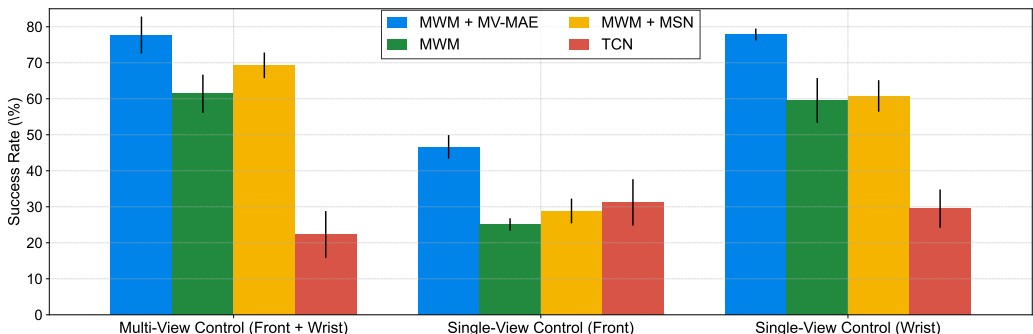

Figure 5: Aggregate success rate of multi-view visual control and single-view visual control agents on six visual manipulation tasks from RLBench (James et al., 2020). We find that MV-MAE consistently outperforms both single-view and multi-view baselines across all setups. Importantly, single-view control agents trained upon multi-view representations from MV-MAE achieve superior performance to other approaches, demonstrating the importance of data diversity. The result shows the mean and standard deviation averaged over 24 runs. We provide the learning curve for all tasks in Appendix F.

- Time Contrastive Network (TCN; Sermanet et al. 2018): TCN learns visual representations through contrastive learning that utilizes observations from simultaneous viewpoints as positive and frames from the same view but taken from different times as negative. We consider TCN as a baseline to evaluate our approach against the approach that enforces view-invariance via contrastive loss requiring a complex scheme for sampling positive-negative pairs.

- MWM + MV-MAE (Ours): We build our approach upon MWM by learning a world model on top of frozen representations from MV-MAE. This differs from MWM in that we train the video autoencoder with view-masking to learn cross-view information from multi-view data.

## 4.2 QUANTITATIVE RESULTS

**Multi-view control with front and wrist cameras.** We first evaluate the effectiveness of our multi-view representation learning scheme in a multi-view visual control setup, where the agent operates on both front and wrist cameras. For all baselines, we extract features from each viewpoint and use concatenated features as inputs to the agent. For our approach, we use generic representations from a single multi-view encoder as inputs. In Figure 5, we first observe that MWM + MV-MAE and MWM + MSN outperform MWM, which shows the importance of multi-view representation learning. However, we find that TCN significantly fails to solve most of the tasks. This shows the critical drawback of TCN, which suffers from mode collapse when negative samples are too similar (*e.g.,* wrist camera observations look similar after grasping the objects in our case). We also find that MWM + MV-MAE outperforms both multi-view baselines, demonstrating that masked view reconstruction can effectively leverage multi-view data for representation learning compared to other approaches that enforce view-invariance despite its simplicity.

**Single-view control.** We also report the performance of single-view visual control agents in Figure 5. For single-view control experiments, we learn visual representations using multi-view data consisting of front and wrist camera observations and train the RL agent that operates on either front or wrist camera. Interestingly, we find MWM + MV-MAE significantly outperforms MWM in both setups, which shows that multi-view representation learning can also be effective when we only have access to single-view data for behavior learning. This result suggests that our approach can be useful for practical scenarios where we can utilize additional third-person cameras during training, but the robot should operate on a single wrist camera at deployment time.

**Three camera experiments.** One limitation of multi-view baselines is that it is difficult to scale up them into a setup with more than two cameras because they learn representations by matching the representation of two views. Meanwhile, our approach is naturally applicable to a setup with more than two cameras because our approach simply reconstructs multi-view data from all cameras. To

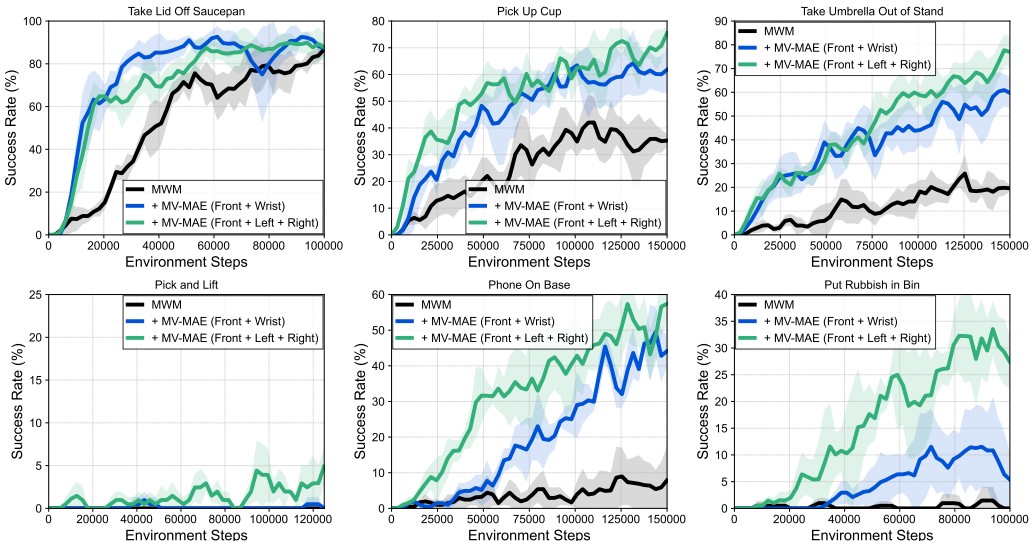

Figure 6: Learning curves of single-view visual control agents that operate on the front camera for solving six visual manipulation tasks from RLBench (James et al., 2020) as measured on the success rate. Front + Wrist and Front + Left + Right in legends denote the combination of cameras used for multi-view representation learning. The solid line and shaded regions represent the mean and standard deviations, respectively, across four runs.

empirically demonstrate this, we report the performance of MV-MAE with three cameras, *i.e.*, front camera, left shoulder camera, and right shoulder camera (see Appendix E for examples of camera observations). We consider this setup motivated by the usage of multiple third-person cameras in prior works (Akkaya et al., 2019; Akinola et al., 2020). Figure 6 shows the performance of representation learning with the three cameras, where we find that MV-MAE with the three cameras outperforms MV-MAE with front and wrist cameras. This shows that the model can capture fine-grained details from three third-person cameras without using the wrist camera, which provides a closer look at the objects. However, we find that several RLBench tasks are still very challenging to solve only using the front camera for behavior learning, *e.g.*, success rates of single-view control agents on Pick and Lift are very low, which necessitates follow-up works on representation learning for visual control. Given these results, a large-scale investigation into the effect of camera configurations on representation learning for visual control could be an interesting future direction.

### 4.3 Ablation study and analysis

**Effect of view-masking.** To investigate the effect of the proposed view-masking scheme, we report the performance with and without view-masking in Figure 7(a). We find that utilizing the view-masking strategy for training the video autoencoder achieves significantly better performance than uniform-masking, which supports that view-masking encourages the autoencoder to learn representations capturing useful cross-view information. On the other hand, we observe that view-masking can be harmful when training the image autoencoder unlike training the video autoencoder. This is because masked view reconstruction could be a challenging task for the image autoencoder and thus prevent the model from learning useful representations, as we mentioned in Section 3.1.

**Effect of video autoencoding.** In Figure 7(a), we also investigate the effect of video autoencoding by reporting the performance of MV-MAE with and without video autoencoding. We find that performance significantly degrades without video autoencoding, which demonstrates that enabling the model to have access to unmasked frames of the same view is indeed crucial for representation learning with masked view reconstruction. However, we find that training the video autoencoder is not effective without the view-masking scheme, achieving even worse performance than training the image autoencoder. This shows that the model can learn to exploit additional information from multiple frames for solving reconstruction tasks, making it difficult to learn useful representations.

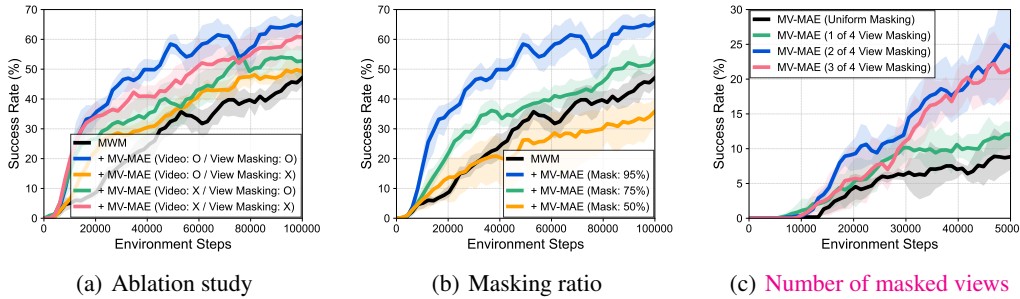

|  | (a) Ablation study | (b) Masking ratio | (c) Number of masked views |

Figure 7: Learning curves of single-view visual control agents operating on the front camera for solving three manipulation tasks from RLBench (James et al., 2020), investigating the effect of (a) view masking and video autoencoding and (b) masking ratio. (c) We report the performance of MV-MAE trained with four cameras (*i.e.,* front, wrist, left shoulder, and right shoulder) and varying number of masked views (*i.e.,* 0, 1, 2, and 3). The solid line and shaded regions represent the mean and stratified bootstrap confidence interval across 12 runs.

**Masking ratio.** In Figure 7(b), we find that performance of MV-MAE keeps increasing with a higher masking ratio, achieving the best performance with an extremely high masking ratio of 95%. We hypothesize this is because spatial information redundancy (He et al., 2021) is more significant in camera observations from visual manipulation tasks than natural images. This also aligns with the observation of prior works (Feichtenhofer et al., 2022; Tong et al., 2022) where the high masking ratio of 90% has shown to be effective for videos with more information redundancy.

**Number of masked views.** In Figure 7(c), we further investigate how the proposed view-masking scheme works with the different number of masked views. Specifically, we report the performance of MV-MAE with four cameras (*i.e.,* front, wrist, left shoulder, right shoulder) and varying number of masked views. We observe that masking more views leads to better performance, outperforming a uniform-masking baseline. This shows that the proposed view-masking can prevent the model from exploiting information redundancy and effectively encourage multi-view representation learning.

## 5 DISCUSSION

We have presented Multi-View Masked Autoencoder (MV-MAE), a simple and scalable framework for multi-view representation learning with multiple cameras. Our experimental results demonstrate that multi-view representation learning can significantly outperform both single-view and multi-view representation learning baselines by a large margin on challenging visual manipulation tasks from RLBench (James et al., 2020). These results suggest that data diversity from multiple viewpoints can play a key role in representation learning for visual control. We believe that our work can be useful for various robot learning scenarios with multiple cameras, by providing a framework that can work with different number of cameras for representation learning and behavior learning phases. We hope our work facilitates future research on leveraging multiple cameras for representation learning.

**Limitation and future directions.** One limitation of our approach is that the compute cost of MV-MAE increases quadratically as the number of viewpoints increases. While we find it possible to train a relatively large model thanks to the extremely high masking ratio, it would be interesting to incorporate a more scalable architecture (Jaegle et al., 2021). As we mentioned in Section 4.2, it would also be interesting to consider a large-scale investigation into the effect of diverse camera configurations on representation learning, by conducting pre-training on large-scale robotics datasets consisting of multi-view data (Dasari et al., 2019). Moreover, by combining the idea of MV-MAE and multimodal MAE (Geng et al., 2022), training an RL agent that can operate on any modality or viewpoints is a direction we would like to pursue in future works. Finally, we are keen to evaluate the effectiveness of multi-view representation learning with MV-MAE on real robots.

REPRODUCIBILITY STATEMENT

We provide the implementation details of our approach and baselines in Section 4, Appendix A, and Appendix C. We also provide our source code to reproduce main results in Appendix C.

ETHICS STATEMENT

Learning visual control agents operating on easily accessible cameras can be helpful for various applications, such as factory automation, autonomous driving, and training collaborative robots. However, there could be a scenario where the robot is misused by malicious users, *e.g.,* attackers might inject rewards that teach undesirable behaviors which could be harmful to society. To prevent such abuses, it is important to develop algorithms with an eye towards safety as well as performance.

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

# A    BASELINES

## A.1    MASKED WORLD MODELS

Masked world models (MWM; Seo et al. 2022a) is a visual model-based RL method that iterates the representation learning phase and the behavior learning phase. In the representation learning phase, MWM learns visual representation by reconstructing pixels given masked convolutional features. In the behavior learning phase, policy, along with the world model, is learned on top of frozen representation. We explain the details of each phase as follows.

**Representation learning.**    For representation learning, MWM trains an autoencoder by (i) reconstructing an image with masked convolutional features and (ii) predicting a reward. MWM processes raw input image using a convolution stem, which is convolutional layers followed by a flatten layer. Then, MWM randomly masks convolutional features, and feeds visible features into the ViT encoder. The outputs from the ViT encoder, along with (learnable) mask tokens, are fed into the ViT decoder, which reconstructs the raw image. For the auxiliary reward prediction, one additional learnable mask token is concatenated into the inputs of the ViT decoder, and the corresponding output representation followed by a linear output head is used to predict the reward.

**Behavior learning.**    Once visual representation is learned, MWM trains an RL agent by actor-critic learning scheme using the imaginary latent states from the world model. For the world model, MWM uses a variant of Recurrent State Space Model (RSSM; Hafner et al. 2019). Formally, for a visual task formulated as a partially observable Markov decision process (Sutton & Barto, 2018), which is defined as a tuple $(\mathcal{O}, \mathcal{A}, p, r, \gamma)$, $\mathcal{O}$ is the observation space, $\mathcal{A}$ is the action space, $p(o_t|o_{<t}, a_{<t})$ is the transition dynamics, $r$ is the reward function that maps previous observations and actions to a reward $r_t = r(o_{\leq t}, a_{\leq t})$ and $\gamma \in [0, 1)$ is the discount factor. Let $z_t$ be an output from the ViT encoder for a current observation $o_t$. The RSSM consists of the following learnable components:

$$
\begin{array}{lll}
\text{Representation model:} & s_t \sim q_\theta(s_t|s_{t-1}, a_{t-1}, z_t) & \\
\text{Transition model:} & \hat{s}_t \sim p_\theta(\hat{s}_t|s_{t-1}, a_{t-1}) & \\
\text{Visual representation decoder:} & \hat{z}_t \sim p_\theta(\hat{z}_t|s_t) & \text{(1)} \\
\text{Reward predictor:} & \hat{r}_t \sim p_\theta(\hat{r}_t|s_t), &
\end{array}
$$

The representation model extracts model state $s_t$ from previous model state $s_{t-1}$, previous action $a_{t-1}$, and $z_t$. The transition model predicts future state $\hat{s}_t$ without access to $z_t$. The visual representation model reconstructs $z_t$ to provide learning signal. All model parameters $\theta$ are jointly trained to minimize the negative variational lower bound (Kingma & Welling, 2014). For behavior learning, MWM predicts latent future states and train a stochastic actor and a deterministic critic to maximize the imagined returns (Hafner et al., 2021). We refer to Seo et al. (2022a) for more details.

## A.2    MWM + MASKED SIAMESE NETWORK

MWM + masked siamese network (MSN; Assran et al. 2022) is a baseline that learns visual information via reconstruction with masked convolutional features, but also cross-view information from different viewpoints. Specifically, we extract the representations of one viewpoint from the encoder, and also the representations of another viewpoint from the target momentum encoder. We first use the representations of the encoder as inputs to the ViT decoder that reconstructs raw pixels. And then compute the representation matching loss that minimizes the distance between two representations. We maintain a target encoder that is updated via an exponential moving average of the (anchor) encoder parameters with a target decay rate of 0.999. For a given (anchor) frame, we regard the corresponding frame from another viewpoint as a target frame. For obtaining representations from each viewpoint, we conduct average pooling on token embeddings from the last layer of the encoder. While original MSN utilizes learnable prototypes for obtaining representations, we find it extremely unstable in our setup. So we adopt the simple approach of Grill et al. (2020) that minimizes the distance between L2 normalized representations. We note that this is still called MSN because the main idea of matching the representation of masked viewpoint representations with unmasked viewpoint target representations.

## A.3 Time contrastive network

Time contrastive network (TCN; Sermanet et al. 2018) is a contrastive approach that learns view-invariant representations by attracting the representations of simultaneous viewpoints but making the representations from the same viewpoints be far located. For a given (anchor) frame in one viewpoint, we sample a positive and negative frame as follows. For the positive frame, we use the frame that has the same timestep as the given anchor frame but from another viewpoint. For the negative frame, we choose a frame that is a temporally faraway frame from the same viewpoint. Specifically, we sample a random frame among frames that are at least 30 timesteps away from the anchor frame. After building a triplet of anchor, positive, and negative frame, we train the encoder model with triplet loss (Schroff et al., 2015), which is formulated as follows:

$$\mathcal{L}_{\text{TCN}} = \mathtt{max}(\|f(o^a) - f(o^p)\|_2^2 - \|f(o^a) - f(o^n)\|_2^2 + \alpha, 0), \tag{2}$$

where $o^a, o^p$, and $o^n$ are anchor, positive, and negative frames, respectively, $f(\cdot)$ refers class embeddings from the last layer of ViT encoder and $\alpha$ is margin, which is set to 2. In the control phase, we freeze the encoder and use average pooled token embeddings as an input for the RL agent.

## B  Pseudocode for Multi-View Masked Autoencoder

We consider a camera observation $o_t^v$ where $t \in \{1, ..., T\}$ denotes a video timestep and $v \in \{1, ..., V\}$ denotes a camera viewpoint. For clarity, we let $\mathcal{T} = \{1, ..., T\}$ and $\mathcal{V} = \{1, ..., V\}$. We let $\mathcal{L}^{\mathtt{actor}}$ and $\mathcal{L}^{\mathtt{critic}}$ be the actor and critic loss of MWM (Seo et al., 2022a).

---

**Algorithm 1** Multi-View Masked Autoencoders for Visual Control

---

1: Initialize replay buffer $\mathcal{B} \leftarrow \emptyset$ and parameters of autoencoder $\phi$, actor $\psi$, critic $\xi$
2: **for** each environment step $i$ **do**
3:     // Multi-view representation learning
4:     Sample minibatch $\{o_t^v\} \sim \mathcal{B}$
5:     **for** each video timestep $t$ **do**
6:         Get convolutional features $z_t^{v,\mathtt{conv}} = f_\phi^{\mathtt{conv}}(o_t^v)$ for $v \in \mathcal{V}$
7:         Sample $\{z_t^{v,\mathtt{conv}}\}_{v \in \Omega_t} \sim \mathtt{ViewMask}(\{z_t^{v,\mathtt{conv}}\}_{v \in \mathcal{V}})$, where $\Omega_t$ is a set of unmasked views
8:     **end for**
9:     Collect unmasked viewpoints $\Omega = \bigcup_{t=1}^{T} \Omega_t$
10:     Get reconstructions $\{\hat{o}_t^v\}_{v \in \mathcal{V}} = f_\phi^{\mathtt{decoder}}(f_\phi^{\mathtt{encoder}}(\{z_t^{v,\mathtt{conv}}\}_{v \in \Omega}))$ for $t \in \mathcal{T}$
11:     Update $\phi$ by minimizing $\sum_{t=1}^{T} \sum_{v=1}^{V} \|o_t^v - o_t^v\|_2$

12:     // Actor critic learning
13:     Sample minibatch $\{(o_1^1, ..., o_1^V, a, r)\} \sim \mathcal{B}$
14:     Get convolutional features $\{z_1^{v,\mathtt{conv}}\}_{v \in \mathcal{V}}$
15:     **if** single-view control with $v_s$ **then**
16:         Get single-view representation $h = f_\theta^{\mathtt{encoder}}(\{z_1^{v_s,\mathtt{conv}}\})$
17:     **else**
18:         Get multi-view representation $h = f_\theta^{\mathtt{encoder}}(\{z_1^{v,\mathtt{conv}}\}_{v \in \mathcal{V}})$
19:     **end if**
20:     Update $\psi$ and $\xi$ by minimizing $\mathcal{L}^{\mathtt{actor}}$ and $\mathcal{L}^{\mathtt{critic}}$ with $\{(h, a, r)\}$

21:     // Collect transitions
22:     Get convolutional features $\{z_{1,i}^{v,\mathtt{conv}}\}_{v \in \mathcal{V}}$ from a current observation $\{o_{1,i}^v\}_{v \in \mathcal{V}}$
23:     **if** single-view control with $v_s$ **then**
24:         Get single-view representation $h_i = f_\theta^{\mathtt{encoder}}(\{z_{1,i}^{v_s,\mathtt{conv}}\})$
25:     **else**
26:         Get multi-view representation $h_i = f_\theta^{\mathtt{encoder}}(\{z_{1,i}^{v,\mathtt{conv}}\}_{v \in \mathcal{V}})$
27:     **end if**
28:     Sample action $a_i \sim f_\psi^{\mathtt{actor}}(h_i)$ and collect reward $r_i$ from environment
29:     Store sample to replay buffer $\mathcal{B} \leftarrow \mathcal{B} \cup \{(o_{1,i}^1, ..., o_{1,i}^V, a_i, r_i)\}$
30: **end for**

---

## C  IMPLEMENTATION DETAILS

**Source code.**  Source code for reproducing our experimental results is available at:

https://anonymous.4open.science/r/iclr2023_mvmae

**RLBench details.**  For RLBench experiments, we designed dense rewards for six manipulation tasks used in our experiments. For simple tasks (*i.e.,* Take Lid Off Saucepan, Pick Up Cup, Take Umbrella Out of Umbrella Stand), we design the reward to be the sum of the distance between a gripper and a target object. For more complex tasks that require more long-term behavior (*i.e.,* Pick and Lift, Phone On Base, Put Rubbish in Bin), we first construct some checkpoints where the robot should reach. Then we define the reward to be the sum of the distance between the gripper and the nearest next checkpoint. For disabling rotation, we use a path planner with identity quaternion to force the robot to be in an upright position. Unlike prior approaches that use the path planner (James & Davison, 2022; James et al., 2022; James & Abbeel, 2022a;b) to specify absolute (x, y, z) position, we train the RL agent to output relative change in (x, y, z) position. This is available by using the action mode EndEffectorPoseViaPlanning(absolute_mode=False) in the RLBench. For more details, we refer readers to the source code we have attached.

**Architecture and optimization details.**  Our architecture is based on the publicly available source code of Seo et al. (2022a), which is implemented with tfimm[2] library. For our method and all baselines, we use the same architecture consisting of 8-layer ViT encoder and 6-layer ViT decoder. For each view and time step, we introduce additional 1D learnable parameters that have the same embedding size as transformer blocks. We add these parameters to 2D fixed sin-cos embeddings and add them to features. We note that these parameters are shared across the same times and the same views. For optimization, we use Adam optimizer (Kingma & Ba, 2015) with the learning rate of $3e - 4$, the weight decay of $1e - 6$, and the batch size of 1024. For training MV-MAE, we apply warm-up learning rate scheduling over initial 2500 gradient steps from learning rate of 0. We take 1 gradient step per every 2 environment steps. We follow the training schemes and details of Seo et al. (2022a) regarding the architecture, unless otherwise specified.

**Computation.**  We use 8 CPU cores (Intel Xeon Gold 6226R @ 3.9GHZ) and 1 GPU (NVIDIA GeForce RTX 3090) for our experiments. We find that there is no significant difference between all methods with regard to wall time for experiments because there is a severe bottleneck from the rendering speed of the RLBench simulator. Running experiments over 150k environment steps takes approximately 18 hours for all methods.

---

[2]https://github.com/martinsbruveris/tensorflow-image-models

**Hyperparameters.** We report the hyperparameters used in our experiments in Table 1.

Table 1: Hyperparameters used in our experiments. Unless otherwise specified, we use the same hyperparameters used in MWM (Seo et al., 2022a).

| Hyperparameter | Value |
|---|---|
| *Representation learning* | |
| Image observation | $96 \times 96 \times 3$ |
| Image normalization | Mean: $(0.485, 0.456, 0.406)$, Std: $(0.229, 0.224, 0.225)$ |
| Autoencoder batch size | 1024 |
| Autoencoder initialization steps | 10000 |
| Autoencoder warm-up steps | 2500 |
| Autoencoder learning rate | $3 \cdot 10^{-4}$ |
| Autoencoder masking ratio | 0.95 (multi-view), 0.9 (single-view) |
| Autoencoder ViT encoder size | 8 layers, 4 heads, 256 units |
| Autoencoder ViT decoder size | 6 layers, 4 heads, 256 units |
| *Behavior learning (MWM)* | |
| Action repeat | 1 |
| Max episode length | 150 |
| Early episode termination | True (when path planner fails) |
| Reward normalization | True |
| World model batch size | 50 |
| World model sequence length | 50 |
| World model tradeoff ($\beta$) | 1.0 |
| World model tradeoff free-bits | 0.1 |
| World model ViT encoder size | 2 layers, 4 heads, 128 units |
| World model ViT decoder size | 2 layers, 4 heads, 128 units |

# D  TWO-CAMERA EXAMPLES

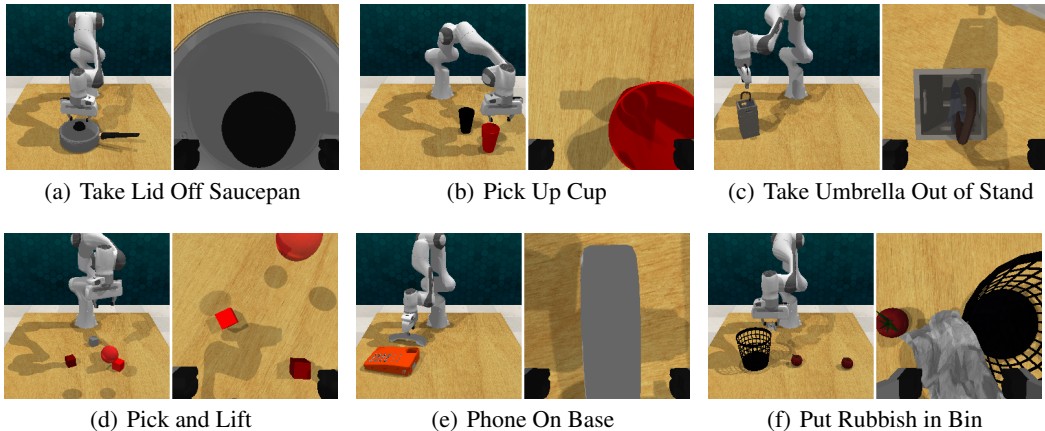

(a) Take Lid Off Saucepan   (b) Pick Up Cup   (c) Take Umbrella Out of Stand

(d) Pick and Lift   (e) Phone On Base   (f) Put Rubbish in Bin

Figure 8: Examples of multi-view data consisting front and wrist camera observations used in our RLBench (James et al., 2020) experiments. Front camera observations provide a broad look at a robot workspace, and wrist camera observations provide a closer look at target objects.

# E  THREE-CAMERA EXAMPLES

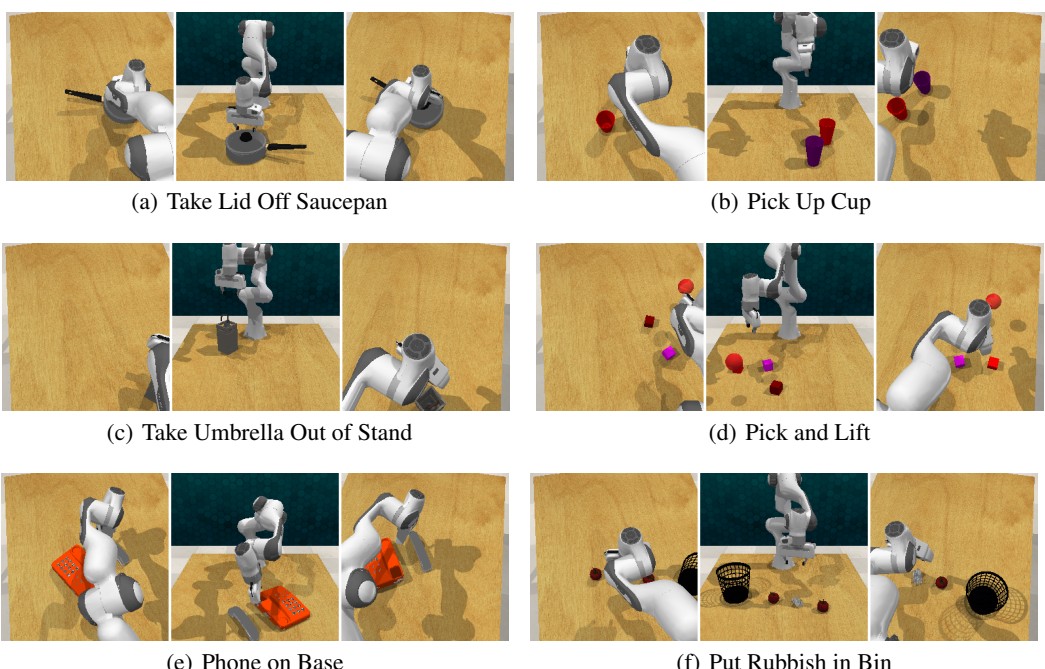

(a) Take Lid Off Saucepan   (b) Pick Up Cup

(c) Take Umbrella Out of Stand   (d) Pick and Lift

(e) Phone on Base   (f) Put Rubbish in Bin

Figure 9: Examples of multi-view data consisting left shoulder, front, and right shoulder observations used in our RLBench (James et al., 2020) experiments.

# F FULL MULTI-VIEW EXPERIMENTAL RESULTS

## F.1 SINGLE-VIEW VISUAL CONTROL WITH FRONT CAMERA

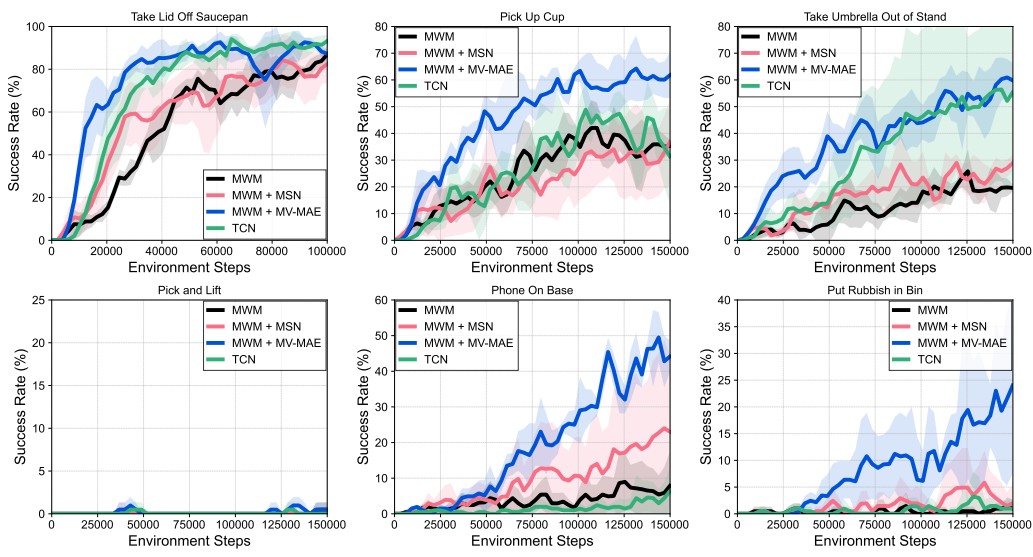

Figure 10: Learning curves of RL agents that operate on front camera observations for solving six visual manipulation tasks from RLBench as measured on the success rate. The solid line and shaded regions represent the mean and standard deviations, respectively, across four runs.

## F.2 SINGLE-VIEW VISUAL CONTROL WITH WRIST CAMERA

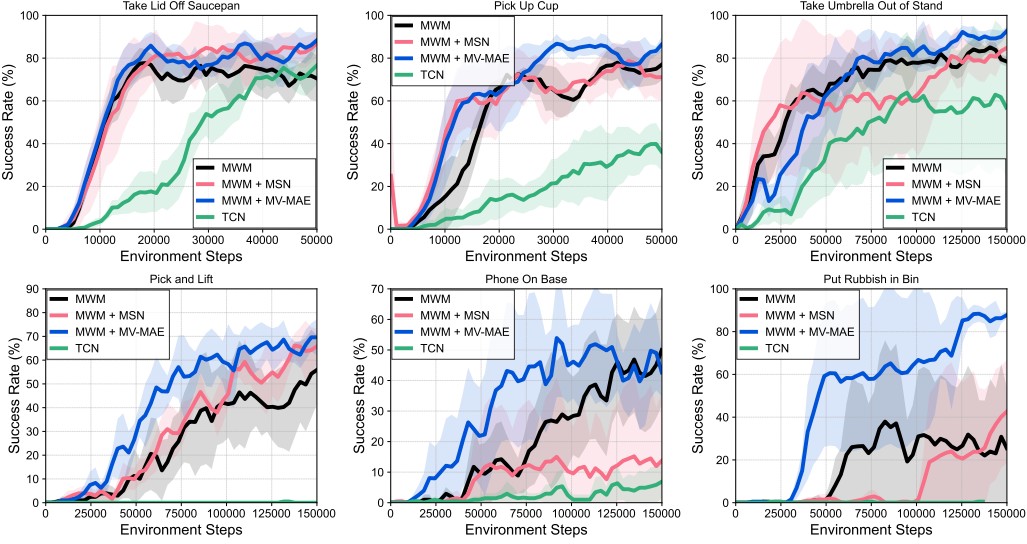

Figure 11: Learning curves of RL agents that operate on wrist camera observations for solving six visual manipulation tasks from RLBench as measured on the success rate. The solid line and shaded regions represent the mean and standard deviations, respectively, across four runs.

## F.3 MULTI-VIEW VISUAL CONTROL WITH FRONT AND WRIST CAMERA

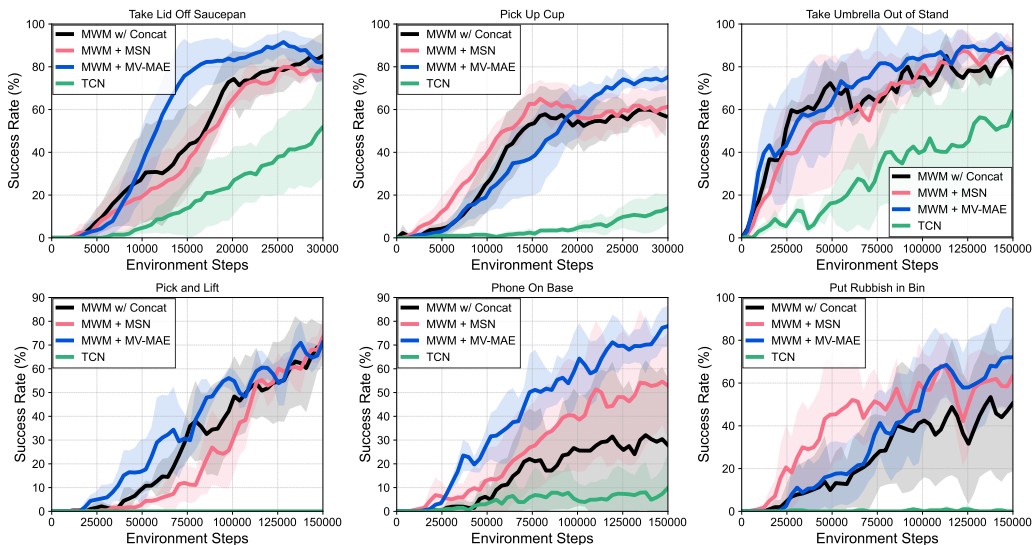

Figure 12: Learning curves of RL agents that operate on front and wrist camera observations for solving six visual manipulation tasks from RLBench as measured on the success rate. The solid line and shaded regions represent the mean and standard deviations, respectively, across four runs.

## G ADDITIONAL EXPERIMENTS

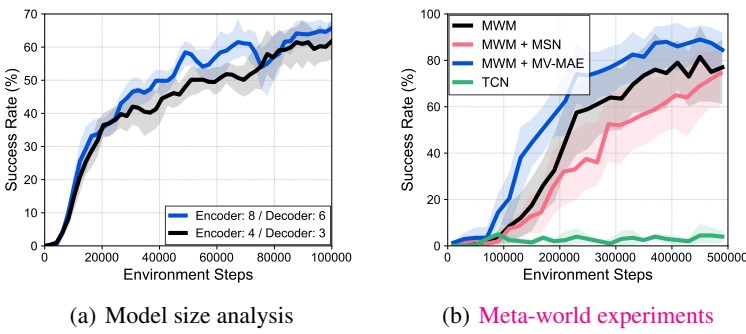

(a) Model size analysis          (b) Meta-world experiments

Figure 13: (a) Learning curves of single-view visual control agents that operate on the front camera for solving visual manipulation tasks from RLBench (James et al., 2020), investigating the effect of model size. (b) Learning curves of single-view visual control agents that operate on the wrist camera for solving visual manipulation tasks from Meta-world (Yu et al., 2020). The solid line and shaded regions represent the mean and stratified bootstrap confidence interval across (a) 12 and (b) 20 runs.

**Model size.** In Figure 13(a), we investigate the effect of scaling up MV-MAE and find that training larger model for visual representation learning with multi-view data improves sample-efficiency. Further scaling up the model and pre-training MV-MAE on large real-world robotics dataset (Dasari et al., 2019) containing multi-view data would be an interesting future work.

**Meta-world experiments.** In order to verify that our approach can also work for different benchmark tasks other than RLBench, we provide additional experimental results on five visual manipulation tasks[3] from Meta-world (Yu et al., 2020) benchmark. Specifically, we train the single-view

---

[3] We report the experimental results on Basketball, Shelf Place, Pick Out Of Hole, Pick Place, and Soccer.

control agent operating on the wrist camera with multi-view representations learned using the front and wrist cameras. For the front camera, we use the third-person camera viewpoint used in Seo et al. (2022a). For the wrist camera, we use the wrist camera introduced in Hsu et al. (2022). For training the MWM agent, we use the same hyperparameters used in Seo et al. (2022a), *e.g.,* mask ratio of 75%, image resolution of 64×64, and taking one gradient step per every five environment steps. For training MV-MAE, we use 95% masking ratio which is the same ratio as used in RLBench experiments. In Figure 13(b), we find that MWM + MV-MAE exhibits better sample-efficiency than MWM, which again shows the benefit of multi-view representation learning. Moreover, we find that baselines struggle to outperform the single-view baseline MWM, and TCN completely fails with the wrist camera similar to the observation in RLBench experiments (see Figure 11). This again demonstrates the wide applicability of our approach to diverse tasks with diverse type of cameras.

**Embedding analysis.**    We also visually demonstrate how representations are learned differently for MV-MAE and TCN that enforces view-invariance in Figure 14. Specifically, we train multi-view agents using MV-MAE and TCN for solving Pick Up Cup task and extract the representations from the front and wrist camera using the visual encoders. Then we visualize the representations using t-SNE (Van der Maaten & Hinton, 2008). We observe that MV-MAE representations of the front and wrist cameras are disentangled from each other, but one can see that representations from both cameras have similar representation structure. This is because we encourage the model to learn cross-view information while also learn the information within each viewpoint simultaneously. On the other hand, front and wrist camera representations from TCN are entangled, which shows how qualitatively MV-MAE and TCN learn different representations.

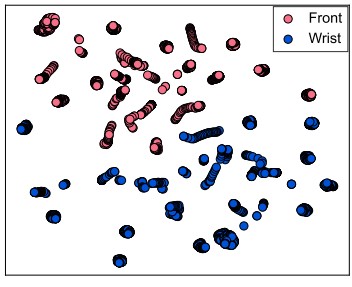
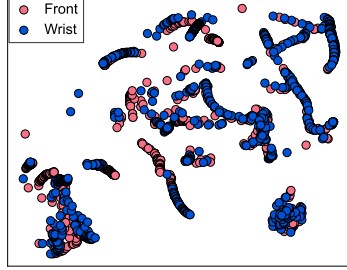

(a) MV-MAE representations            (b) TCN representations

Figure 14: t-SNE (Van der Maaten & Hinton, 2008) visualization of representations extracted from front and wrist camera observations from (a) MV-MAE and (b) TCN.

