# OpenReview forum: "Multi-View Masked Autoencoders for Visual Control"
_ICLR.cc/2023/Conference — Submitted to ICLR 2023_

### Official Review · Reviewer_HaAG · 2022-10-20

**Confidence:** 4
**Correctness:** 4
**Technical Novelty And Significance:** 2
**Empirical Novelty And Significance:** 2
**Recommendation:** 5

**Clarity, Quality, Novelty And Reproducibility:**

- Easy to follow. Well written.

- Likely reproducible.

- The work appears to be original.

**Strength And Weaknesses:**

Strengths

- Interesting problem area.

- Well written and easy to follow.

- Evaluated versus baselines, including an ablation study.

Weaknesses

- Lack of technical novelty

The proposed MV-MAE is an agglomeration of recent techniques. The underlying method is MAE [1], but using input in the style of [2] (convolutional features, instead of raw pixels). A learnable parameter reflecting the camera viewpoint / frame timestep is added to each token in the style of [3]. The rest follows a standard video masked autoencoder [4]. Though a view masking strategy is proposed, it is ultimately not compared against any alternative strategies. The visual control portion of the paper uses representations from MV-MAE as input to an off-the-shelf RL algorithm from [2] with evaluation from benchmark suite RLBench.

[1] He, Kaiming, et al. "Masked autoencoders are scalable vision learners." IEEE/CVF Conference on Computer Vision and Pattern Recognition. 2022.

[2] Younggyo Seo, Danijar Hafner, Hao Liu, Fangchen Liu, Stephen James, Kimin Lee, and Pieter Abbeel. "Masked world models for visual control." Conference on Robot Learning, 2022a.

[3] Geng, Xinyang, et al. "Multimodal Masked Autoencoders Learn Transferable Representations." International Conference on Machine Learning: Workshop on Pre-Training, 2022.

[4] Feichtenhofer, Christoph, et al. "Masked Autoencoders As Spatiotemporal Learners." arXiv preprint arXiv:2205.09113 (2022).

- Lack of detail

Most of the pieces of the proposed approach are from pre-existing works, therefore no detail is given in the manuscript outside of the reference. For example, what's the format of the learnable parameter representing camera ID / frame ID and how are they integrated? Additionally, what are the optimization details and how does the iterative training process work?

- Limited evaluation

Though the evaluation is fairly extensive, the experiments don't give insight into specific design choices that are crucial for MV-MAE to "work". For example, what is the impact of the view masking strategy?

**Summary Of The Paper:**

This paper explores multi-view representation learning and introduces the Multi-View Masked Autoencoder (MV-MAE) framework. A video autoencoder is updated to function with multi-view input video streams. The learned representations are shown to be useful for the downstream task of visual control by training reinforcement learning agents. Evaluation is conducted on visual manipulation tasks from RLBench. Results show improvement over baselines.

**Summary Of The Review:**

The topic area, multi-view representation learning, is interesting.  Empirically, the proposed method (MV-MAE) shows improvement over baselines on the downstream task of visual control. However in terms of technical novelty, the representation learning component is a slightly tweaked masked autoencoder. The visual control portion exists just for evaluating the learned representations, and uses an off-the-shelf RL algorithm from a recent paper. The manuscript is of high quality but very little space is allocated to describing the proposed method (essentially just Section 3.1). For me, the paper reduces to applying masked autoencoders to a pre-existing visual control task. The results are compelling, but novelty is lacking, which makes this a tough paper to review.

Ultimately, I feel this paper falls on the "marginally below the acceptance threshold" side of things. For me, I would want the paper to allocate more space to method description: highlighting important details that make it work (e.g., is it the specific choice of view masking strategy?) and adding experiments that reflect why this is the right way to structure things in the multi-view case.

---

> ### Author Response · Authors · 2022-11-13
> **Response to Reviewer HaAG**
>
> Dear reviewer HaAG,
>
> We sincerely appreciate your valuable comments. We found them extremely helpful in improving our draft. We address each comment in detail, one by one below.
>
> ---
>
> **Q1: Technical novelty of MV-MAE**
>
> **A1.** We emphasize that our work considers a novel and practical setup of (i) multi-view representation learning with multiple cameras of diverse types and (ii) training RL agents with different numbers of cameras for representation learning and behavior learning. Moreover, to the best of our knowledge, we are first to propose a recipe for successfully training MAE with multi-view data. While MAE has been effective for representation learning from images or videos, how to extend MAE into a multi-view setup has not yet been investigated. Specifically, our work introduces a new view-masking strategy for training MAE with multi-view data to encourage multi-view representation learning and provides experimental results that show its effectiveness over a uniform-masking strategy (in Figure 7(a) of the original draft). We also provide an analysis of the effect of mask ratio and the number of masked viewpoints, which could be helpful for future researchers. Furthermore, our work demonstrates that a combination of view-masking and video masked autoencoding can be synergistic for multi-view representation learning. We think our effective yet simple multi-view representation learning scheme and extensive experiments on the novel setup can serve as an important step toward sample-efficient robot learning.
>
> ---
>
> **Q2: Lack of detail**
>
> **A2.** Thanks for pointing this out.
> We have included the details of learnable parameters used in MV-MAE in the revised draft.
> We have included optimization details in Appendix B of the revised draft.
> We have updated Figure 3 to illustrate our iterative training procedure and included a pseudocode of our framework in Appendix B of the revised draft.
>
> ---
>
> **Q3. Limited evaluation on design choices**
>
> **A3.** Thanks for your suggestion to provide an analysis of the design choices of MV-MAE. We first remark that we already have provided analysis on the impact of the proposed view-masking scheme over uniform masking along with the effect of video autoencoding (Figure 7(a)) and the effect of masking ratio (Figure 7(b)) in Section 4.3 of the original draft. To further address your concern, we provide additional analysis on how the number of masked viewpoints affects the downstream performance in Figure 7(c) of the revised draft. It would be very appreciated if you could let us know which additional analysis could further strengthen the understanding of our approach.

---

> > ### Comment · Reviewer_HaAG · 2022-11-29
> > **Response to Authors**
> >
> > Thanks for the response. I've read the other reviews, the rebuttal, and the updated manuscript.
> >
> > It appears the other reviewers are in agreement that the largest weakness of this work is the limited technical novelty. To recap, the primary contribution of this work is a method for multi-view representation learning (MV-MAE). The learned representations are then showcased for a downstream task in visual control (using the framework of Seo et al. 2022a, with an iterative training scheme).
> >
> > As indicated in my initial review, the proposed method reduces to a slightly modified video masked autoencoder and only a couple of paragraphs in the manuscript are dedicated to method description (Section 3.1). The authors seem to agree with this assessment based on the provided response ("effective yet simple multi-view representation learning scheme"). Evaluation for the downstream task (training an RL agent for visual control) follows pre-existing work and serves only to demonstrate the usefulness of the learned representations.
> >
> > Therefore, for me, the manuscript allocates far too little space for the proposed method (extending video MAE to multi-view, Section 3.1) and far too much space for the downstream task (visual control, essentially Section 3.2 onward). The evaluation largely focuses on demonstrating that the proposed method is better than baselines for the downstream task, as opposed to highlighting why components of the proposed method lead to the correct way for extending video masked autoencoders to multiple views. View masking is perhaps the most crucial component for MV-MAE, and only one experiment is dedicated to this (Section 4.3).
> >
> > Though the authors did improve the manuscript and add additional detail in the revision, I stand with my initial assessment. The limited technical novelty and majority focus of the manuscript on a single pre-existing task lead me to a rating of marginally below the acceptance threshold.

---

> > > ### Author Response · Authors · 2022-12-12
> > > **Response to Reviewer HaAG**
> > >
> > > Dear Reviewer HaAG, we respond to your follow-up comments as follows.
> > >
> > > **Q1. Technical novelty**
> > >
> > > **A1.** As highlighted by Reviewer Zds9, the technical novelty of our work lies in first introducing a recipe for successfully training MAE with multi-view data for visual control. We would like to also remark that being simple to describe could be another strength of our approach without introducing redundant complexities that make our method work. Moreover, we emphasize that we have provided experiments that investigate (i) the effect of view-masking with and without video auto encoding (Fig 7-a), (ii) the effect of masking ratio when used with view-masking (Fig 7-b), and (iii) the effect of the number of masked viewpoints (Fig 7-c).
> > >
> > > ---
> > >
> > > **Q2. Evaluation on a single pre-existing task**
> > >
> > > **A2.** We emphasize that our experiments are designed to carefully evaluate the benefit of our method under novel setups that can be practical and important for robotics. For instance, we have shown that prior works that view-invariance can struggle when we use a widely-used camera configuration of front and wrist cameras, while our method can effectively utilize such multi-view data from different cameras. We also showed that we can use different number of cameras for representation learning and behavior learning, which could be very practical for robotics where fast inference is important. As we stated in previous response, it would be very appreciated if you could suggest which experiments could further strengthen our draft.

---

### Official Review · Reviewer_ntdx · 2022-10-23

**Confidence:** 4
**Correctness:** 4
**Technical Novelty And Significance:** 2
**Empirical Novelty And Significance:** Not applicable
**Recommendation:** 5

**Clarity, Quality, Novelty And Reproducibility:**

__Clarity__: The paper is clearly written and easy to follow.

__Quality__: The quality of writing and visualization are both good.

__Novelty__: The technical novelty is marginal. The paper uses existing architecture and applies it into a new setting.

__Reproducibility__: The paper seems to provide enough details to reproduce the results. But I did not try it myself.


**Strength And Weaknesses:**

__Strengths:__
- The paper writing is in general clear and easy to follow.
- The performance gain is significant compared with single-view and other multi-view baselines. Ablation experiments are also on point and validate various design choices
-  Detailed appendix about reproducibility.


__Major Weaknesses:__
- "Method" part is argurably the weakest part of the paper, as this work pretty much follows the _Masked World Models for Visual Control_ paper and extend it into a cross-view setting without major changes. It applies MAE into the multi-view scenario and validates its performance in this setting.
- The message this work delivers is well-known already. It is obvious that learning from cross-view cameras can boost the single-view results.
- Some minor issues regarding the paper writing, typo and citations. (See below)

__Minor Issues:__
- First paragraph of introduction. starting off the paper with multi-modal learning is inappropriate. The paper is using camera only and it's not comparing itself with nor improving upon multi-modal approaches. Quoting multi-view camera papers makes more sense in backing up your argument in diversity, as opposed to quoting multi-modal papers.
- Do not include Google Brain in the citation of "Time-contrastive networks: Self-supervised learning from video"
- Typo. Figure 1 caption, the autoencoder reconstruct"s" all frames at the same time

**Summary Of The Paper:**

Motivated by the fact that data diversity plays a key role in representation learning, this paper proposes "Multi-View Masked Autoencoder" to leverage the cross-view and cross-frame information to learn the visual representation for control tasks. Experiments validate that the proposed multi-view representation learning framework outperforms both single-view and other multi-view representation learning baselines.

**Summary Of The Review:**

The paper is well written and it has its significance in experimental results as the improvement over baselines is noticable and the ablation study is well executed. But the technical novelty is limited as it directly applies existing Visual Control MAE model into the cross-view scenario. The message about the importance of using multi-view information is delivered by many other vision work already so it's not new as well. Given the great results and the marginal novelty, the paper straddles right on the acceptance threshold, leaning a bit towards rejection due to the lack of new messages.

---

> ### Author Response · Authors · 2022-11-13
> **Response to Reviewer ntdx**
>
> Dear reviewer ntdx,
>
> We sincerely appreciate your valuable comments. We found them extremely helpful in improving our draft. We address each comment in detail, one by one below.
>
> ---
>
> **Q1: It is obvious that learning from multi-view data can boost single-view performance**
>
> **A1.** We would like to remark that utilizing multi-view data does not always lead to better performance and careful design for multi-view representation learning is important. As you pointed out, learning from multi-view data has been investigated in prior work. However, our experiments show that baseline methods (i.e., TCN and MSN)  that enforce invariance between viewpoints struggle to outperform a single-view baseline under our setup that utilize front and wrist cameras. We expect this is because these cameras are very different from each other, breaking the assumption of baselines that viewpoints have similar characteristics so that view-invariance can be enforced. On the other hand, we show that MV-MAE can effectively learn useful representations from 2, 3, or 4 cameras of diverse types, consistently outperforming the single-view baseline.
>
> ---
>
> **Q2: Technical novelty of MV-MAE**
>
> **A2.** As we mentioned in A1, we emphasize that our work considers a novel setup of (i) multi-view representation learning with multiple cameras of diverse types and (ii) training RL agents with different numbers of cameras for representation learning and behavior learning. Moreover, to the best of our knowledge, we are first to propose a recipe for successfully training MAE with multi-view data. While MAE has been effective for representation learning from images or videos, how to extend MAE into a multi-view setup has not yet been investigated. Specifically, our work introduces a new view-masking strategy for training MAE with multi-view data to encourage multi-view representation learning and provides experimental results that show its effectiveness over a uniform-masking strategy. We also provide an analysis of the effect of mask ratio and the number of masked viewpoints, which could be helpful for future researchers. Furthermore, our work demonstrates that a combination of view-masking and video masked autoencoding can be synergistic for multi-view representation learning. We think our effective yet simple multi-view representation learning scheme and extensive experiments on the novel setup can serve as an important step toward sample-efficient robot learning.
>
> ---
>
> **Q3: Editorial comments on paper writing, typo and citations**
>
> **A3.** Thanks for your helpful comments. We incorporated the comments as follows:
> We re-write the first paragraph of the introduction to discuss the importance of data diversity with less emphasis on multimodal learning approaches.
> We removed Google Brain from the reference of [Sermanet et al., 2018].
> We fixed the typo in the caption of Figure 1.
>
> [Sermanet et al., 2018] Sermanet, Pierre, Corey Lynch, Yevgen Chebotar, Jasmine Hsu, Eric Jang, Stefan Schaal, and Sergey Levine. "Time-contrastive networks: Self-supervised learning from video." ICRA 2018

---

> > ### Comment · Reviewer_ntdx · 2022-11-29
> > **Response to Authors**
> >
> > Thanks for your reply. I have carefully checked the rebuttal and the comments from other reviewers. The authors addressed some of my comments about the editorial suggestion and the multi-view vs. single-view message. However, I'm not fully convinced by the reply from the authors regarding the meaningful message delivered by this work. There isn't a strong motivation behind proposing a new architecture instead of adapting prior work (*e.g.*, *Masked Siamese Network* and *Time Contrastive Network*) to the RL setting. The authors also view their work as *"extending MAE into a multi-view setup"*, which is more of an engineering task than a task with originality. Therefore, I choose to keep my current score.

---

> > > ### Author Response · Authors · 2022-12-12
> > > **Response to Reviewer ntdx**
> > >
> > > Dear Reviewer ntdx, we respond to your follow-up comments as follows.
> > >
> > > **Q1. Motivation for MV-MAE**
> > >
> > > **A1.** The main motivation for introducing MV-MAE is to allow for effectively leveraging multi-view data from diverse cameras for representation learning. As we stated in Related Work, prior works on utilizing multi-view data rather focused on using them as additional inputs or assumed a setup where cameras have different characteristics. Unlike these works, we showed that (i) we can leverage the different number of cameras for representation learning and behavior learning with MV-MAE, and (ii) MV-MAE can effectively learn useful multi-view representations from different types of cameras. Thus we believe the extension of MAE to a multi-view setup is not just an engineering task but could be a valuable contribution to the community by introducing a method that can be useful for practical and novel scenarios for visual control.

---

### Official Review · Reviewer_Zds9 · 2022-10-25

**Confidence:** 3
**Correctness:** 4
**Technical Novelty And Significance:** 3
**Empirical Novelty And Significance:** 3
**Recommendation:** 6

**Clarity, Quality, Novelty And Reproducibility:**

* The proposed approach is novel.
* The experiments are solid and the author(s) provide code (assume to be reproducible).
* The paper is written in high clarity and easy to read.

**Details Of Ethics Concerns:**

* The author(s) mention in their statement about the possibility of misuse in robot and recommend safety use of the approach. This is applied to all research related to robotics not just this paper.

**Strength And Weaknesses:**

# Strength
- The idea of multi-view representation learning with MAE and applied to visual control tasks is novel and interesting.
- Experimental results are strong and superior to baselines. The set of selected baselines fully cover most cases of comparisons(to understand the differences). Ablation experiments also fully cover most of design choices.
- Written presentation is great (high clarity and easy to follow).

# Weakness
- Experiments are done on only one benchmarks.
- On the downstream application of visual control tasks, only one model, aka MWM, is used with MV-MAE.

**Summary Of The Paper:**

This paper presents Multi-View Masked Autoencoder (MV-MAE) for multi-view representation learning and show its benefits in visual manipulation tasks. Experiments are conducted on RLBench demonstrating that MV-MAE (with MWM) significantly outperforms baselines (MWM, MWM+MSN, TCN) consistently across various setups. Ablation experiments provide enough details to understand the effects of different design choices. Written presentation is clear and easy to read & understand.

**Summary Of The Review:**

As mentioned above, the proposed approach is novel and experimental results solid, I would recommend an acceptance for this paper.

---

> ### Author Response · Authors · 2022-11-13
> **Response to Reviewer Zds9**
>
> Dear reviewer Zds9,
>
> We sincerely appreciate your valuable comments. We found them extremely helpful in improving our draft. We address each comment in detail, one by one below.
>
> ---
>
> **Q1: Additional benchmark**
>
> **A1.** Following your suggestion, we additionally report the performance of MV-MAE on several visual manipulation tasks from Meta-world [Yu et al., 2020] benchmark in Appendix G of the revised draft. We find that our approach can also improve the performance of the single-view control agent with multi-view representation learning on Meta-world benchmark, which shows that our approach is generally applicable to diverse manipulation tasks.
>
> [Yu et al., 2020] Yu, Tianhe, Deirdre Quillen, Zhanpeng He, Ryan Julian, Karol Hausman, Chelsea Finn, and Sergey Levine. "Meta-world: A benchmark and evaluation for multi-task and meta reinforcement learning."CoRL 2020.
>
> ---
>
> **Q2: Additional RL algorithm**
>
> **A2.** We agree with the reviewer in that additional experiments with other RL algorithms could be interesting, and we note that our representation learning method is applicable to any RL algorithm. But we find that it is non-trivial to make other RL algorithms (e.g., DrQ-v2 [Yarats et al., 2022]) work on challenging RLBench benchmark with frozen representations. Nonetheless, we will try to have the results ready during the rebuttal period or for the final draft.
>
> [Yarats et al., 2022] Yarats, Denis, Rob Fergus, Alessandro Lazaric, and Lerrel Pinto. "Mastering visual continuous control: Improved data-augmented reinforcement learning." ICLR 2022.

---

### Official Review · Reviewer_mrqQ · 2022-10-25

**Confidence:** 3
**Correctness:** 4
**Technical Novelty And Significance:** 2
**Empirical Novelty And Significance:** 2
**Recommendation:** 5

**Clarity, Quality, Novelty And Reproducibility:**

**Clarity and Quality**

The proposed framework MV-MAE is technically sound. It's an multi-view extension of prior work MWM (Seo et al. 2022a). Even though simple, MV-MAE works well on RLBench tasks. Each sub-module is properly ablated. Overall, the paper is clean and easy to understand.

**Novelty**

As I stated in the weakness section, the proposed MV-MAE is a bit incremental considering prior work in visual pre-training (MAE, video-MAE) and in pre-training for control (MWM). Also, there are existing work studying the benefit of multi-view data.

**Reproducibility**

The source code has been submitted. Even though I didn't run it and try to reproduce the results in the paper, the code reads reasonable.

**Details Of Ethics Concerns:**

I don't have any ethics concerns so far. However, I didn't check research integrity issues (e.g., plagiarism, dual submission) since I don't know what's a good way to do this.

**Strength And Weaknesses:**

**Strength**

- This paper first verified the effectiveness of using multi-view data for pre-training. Moreover, the multi-view pre-training also helps the single-view control tasks.
- The proposed framework MV-MAE is conceptually simple yet works great. It out-performs the single-view baseline and alternative self-supervised methods baseline. Each sub-modules also contribute to the final performance.
- Experiments are solid and the results are promising.
- Writing is easy to follow. Related work is properly discussed.

**Weakness**

- To me, the problem setting isn't challenging enough to test transfer learning. The pre-training and the testing scene are all RLBench and almost the same (same background, robot, etc). Compared to the normal setting in CV tasks, the training-testing gap isn't significant enough. It's encouraging that the visual pre-training still help in this scenario, but it could be more convincing if such representations can be tested under different setting.
- Even though the masked auto-encoder (MAE) models outperfoms the Masked Siamese Network (MSN) and Time Contrastive Network (TCN), I don't get the motivation behind it. What's the issue that multi-view MAE addressed? What's the scenario that MSN/TCN fail to handle but MAR is able to? What's the difference between the learned features? Can we somehow visualize it and see the difference clearly?
- The technical contribution is somewhat limited as 1) video MAE is proposed in Feichtenhofer et al., 2022, and 2) the benefit of using multi-view data for control has been studied before (related work last paragraph). This work is a bit incremental considering prior work MWM (Seo et al. 2022a).


**Summary Of The Paper:**

This paper designs a multi-view masked autoencoder for learning features for visual control. The motivation is that better visual representation leads to better control, which can be obtained from multi-view pre-training using the proposed framework. Results on RLBench have validated the effectiveness of the proposed framework and each introduced components.

**Summary Of The Review:**

This paper presents a framework for using multi-view visual pre-training for robot control. Even though sound and promising, the technical contribution and the experimental evaluations are both limited.

---

> ### Author Response · Authors · 2022-11-13
> **Response to Reviewer mrqQ**
>
> Dear reviewer mrqQ,
>
> We sincerely appreciate your valuable comments. We found them extremely helpful in improving our draft. We address each comment in detail, one by one below.
>
>
> ---
>
> **Q1: Problem setting isn’t challenging enough to test transfer learning**
>
> **A1.**
> We would like to clarify that our paper is not about transfer learning, but considers a standard reinforcement learning setup where a policy is trained using the samples collected from environment interaction, i.e., training and test domains are the same. Even under this setup, we would like to note that learning visutal representations for visual RL is very challenging and has been actively studied [Jaderberg et al., 2017; Srinivas et al., 2020; Hafner et al., 2021; Yarats et al., 2022]. Improving the generalization in RL is definitely an important direction, but this is by far beyond the scope of our work. We have clarified this in the revised draft by updating Figure 3 to reflect that we consider RL setup.
>
> [Jaderberg et al., 2017] Jaderberg, Max, Volodymyr Mnih, Wojciech Marian Czarnecki, Tom Schaul, Joel Z. Leibo, David Silver, and Koray Kavukcuoglu. "Reinforcement learning with unsupervised auxiliary tasks." ICLR 2017
>
> [Srinivas et al., 2020] Srinivas, Aravind, Michael Laskin, and Pieter Abbeel. "Curl: Contrastive unsupervised representations for reinforcement learning." ICML 2020
>
> [Hafner et al., 2021] Hafner, Danijar, Timothy Lillicrap, Mohammad Norouzi, and Jimmy Ba. "Mastering atari with discrete world models." ICLR 2021
>
> [Yarats et al., 2022] Yarats, Denis, Rob Fergus, Alessandro Lazaric, and Lerrel Pinto. "Mastering visual continuous control: Improved data-augmented reinforcement learning." ICLR 2022.
>
> ---
>
> **Q2: Motivation behind MSN, TCN, and MV-MAE**
>
> **A2.** Based on the intuition that extracting shared information from multiple viewpoints can be useful, MSN and TCN enforce invariance between multiple viewpoints. However, this makes MSN and TCN struggle to outperform a single-view baseline when two very different types of cameras (i.e., front and wrist cameras) are given as input in our experiments. On the other hand, MV-MAE learns both spatial correspondences between viewpoints and spatial information within each viewpoint. This enables MV-MAE to consistently outperform the single-view baseline under our experimental setup with multiple cameras of diverse types, in contrast to baseline methods.
>
> Following your suggestion, we also visualize how MV-MAE and TCN representations differ in Figure 14 of the revised draft. The visualization shows that TCN representations from both front and wrist cameras are entangled, i.e., invariant to each other, while MV-MAE representations from each viewpoint are disentangled.
>
> ---
>
> **Q3: Technical novelty of MV-MAE**
>
> **A3.** As we mentioned in A2, we emphasize that our work considers a novel and practical setup of (i) multi-view representation learning with multiple cameras of diverse types and (ii) training RL agents with different numbers of cameras for representation learning and behavior learning. Moreover, to the best of our knowledge, we are first to propose a recipe for successfully training MAE with multi-view data. While MAE has been effective for representation learning from images or videos, how to extend MAE into a multi-view setup has not yet been investigated. Specifically, our work introduces a new view-masking strategy for training MAE with multi-view data to encourage multi-view representation learning and provides experimental results that show its effectiveness over a uniform-masking strategy. We also provide an analysis of the effect of mask ratio and the number of masked viewpoints, which could be helpful for future researchers. Furthermore, our work demonstrates that a combination of view-masking and video masked autoencoding can be synergistic for multi-view representation learning. We think our effective yet simple multi-view representation learning scheme and extensive experiments on the novel setup can serve as an important step toward sample-efficient robot learning.

---

> > ### Comment · Reviewer_mrqQ · 2022-11-29
> > **Response to Authors**
> >
> > Thanks the authors for their reply to my previous comments. I have read it together with the reviewing thread of other reviewers. After reading, my major concerns hold: 1) I understand that this paper is mainly following the problem setting of previous work. However, it is not convincing enough to me and better setting could be introduced and studied as new contribution. 2) As agreed by others, the technical novelty of this work is somewhat limited. It extends the prior work to a multi-view setting. The authors argue that this paper considers a novel and practical setup that using multi-view images from RL agents. I'm fully convinced and to me this is a bit incremental. Most importantly, I don't see the new technical challenges posed by this setup. Both sub-modules have been introduced before and the final framework is more like a A+B solution to me.

---

> > > ### Author Response · Authors · 2022-12-12
> > > **Response to Reviewer mrqQ**
> > >
> > > Dear Reviewer mrqQ, we respond to your follow-up comments as follows.
> > >
> > > **Q1. Problem setting is not convincing**
> > >
> > > **A1.** We would like to emphasize again that our problem setup is a very standard reinforcement learning setup, and this is not a specific setup from some prior works. Moreover, representation learning for RL already requires learning generalizable representations because of the distribution shift between states on which agents operate and the states stored in a replay buffer.  For instance, RL agents should operate on states which could differ from states stored in a replay buffer due to the different environment initializations or the usage of stochastic policies.
> > >
> > > ---
> > >
> > > **Q2. Technical novelty**
> > >
> > > **A2.** We would like to point out that our method is not a naive extension of video masked auto encoding to multi-view setup. Our experiments clearly show that video masked auto encoding with multi-view data can actually be harmful by introducing information redundancy, and proposed view-masking is crucial for making it be effective for multi-view representation learning. As highlighted by Reviewer Zds9, this has not yet been investigated, and presenting a successful recipe for this can be our technical novelty.

---

### Author Response · Authors · 2022-11-13
**General response**

Dear reviewers and AC,

We deeply appreciate your time and effort to review our draft.

Our work proposes a novel combination of a new view-masking strategy and video masked autoencoding for multi-view representation learning. We are delighted to find that all reviewers highlighted our strong empirical performance, extensive ablation study, and clear write-up.

In response to the questions and concerns reviewers raised, we have carefully revised and improved the draft with the following additional experiments and discussions:
- Highlighting our contributions (Section 1)
- Clarification on the message from our experiments (Section 1 and Section 4)
- Clarification on our iterative training procedure for reinforcement learning (Figure 3 in Section 3 and Appendix B)
- Additional details of MV-MAE (Section 3 and Appendix C)
- Additional analysis with varying number of masked viewpoints (Figure 7(c) in Section 4)
- Additional experiments on Meta-world benchmark (Figure 13(b) in Appendix G)
- Visualization of representations learned with MV-MAE (Figure 14 in Appendix G)

These updates are temporarily highlighted in “magenta” for your convenience to check.

We also appreciate your continued effort to provide further feedback until the very end of response/discussion phase. We will make sure to reflect the comments in the final version.

Thank you very much,

Authors.

---

### Decision · Program_Chairs · 2023-01-20

**Decision:**

Reject

**Justification For Why Not Higher Score:**

While the authors improved the submission and made some clarification in the discussion stage, the reviewers were still not convinced and were firm with their initial assessment. By the end of the discussion stage, three out of four reviewers kept their ratings as marginally below the acceptance threshold. The other reviewer was slightly positive about the novelty, but mentioned the pending experiment and low confidence regarding if the current setup is challenging enough.

Given the reviewers’ comments and recommendation, the area chairs did not find sufficient cause to overturn the reviewers' consensus. The area chairs think the paper has great potential and implications for practical applications. By addressing the remaining concerns (e.g., technical novelty and more comprehensive evaluation), the paper would improve a lot in the next cycle.


**Justification For Why Not Lower Score:**

N/A.

**Metareview: Summary, Strengths And Weaknesses:**

Summary of the paper: This paper considers learning a good visual representation for downstream visual manipulation tasks. The key idea is to extend the masked autoencoder to be capable of leveraging multi-view input video streams. Experiments on RLBench show the effectiveness of the proposed method for training a standard reinforcement learning agent, outperforming both single-view and other multi-view representation learning baselines. Also, the proposed method is flexible for training reinforcement learning agents with different numbers of cameras during representation learning and behavior learning. Some ablation studies are performed to investigate the effect of different design choices.

Strengths: The reviewers recognize the major strengths of the paper as 1) a simple yet effective approach; 2) a good set of experiments with strong results and informative ablations for a specific task/setup.

Weaknesses and missing in the submission: The rebuttal has addressed some of the reviewers’ concerns. But there is a general consensus on the weaknesses and limitations of the paper among reviewers, as summarized below:

1) Lack of technical novelty is the major weakness. The proposed method can be viewed as a combination of existing works: extending video masked autoencoders (Feichtenhofer et al., 2022) to multi-view and following standard visual control framework (Seo et al. 2022a) for downstream task evaluation. As mentioned by the reviewers, except this combination, the specific technical contribution of the work is limited and also not well highlighted in the current submission. In particular, as pointed out by the reviewers, why components of the proposed method lead to the correct way for extending video masked autoencoders to the multi-view setting and why the proposed method outperforms baselines - there is a lack of in-depth discussion and empirical validation in the submission.

2) Reviewers are also concerned with the setting of the paper, and argue that more challenging settings should be considered, such as addressing the domain shift between pre-training and testing, conducting additional experiments on other benchmarks, and investigating other reinforcement learning algorithms. The authors followed the reviewers’ suggestion. Some results are provided by the end of the discussion stage (e.g., on the Meta-World benchmark), while other results are pending (making the proposed method work for another reinforcement learning algorithm.